# RAD51 Is Implicated in DNA Damage, Chemoresistance and Immune Dysregulation in Solid Tumors

**DOI:** 10.3390/cancers14225697

**Published:** 2022-11-20

**Authors:** Chengcheng Liao, Srikanth Talluri, Jiangning Zhao, Shidai Mu, Subodh Kumar, Jialan Shi, Leutz Buon, Nikhil C. Munshi, Masood A. Shammas

**Affiliations:** 1Department of Adult Oncology, Harvard (Dana Farber) Cancer Institute, 450 Brookline Ave, Boston, MA 02215, USA; 2VA Health Care System, Boston, MA 02215, USA; 3Department of Hematology/Oncology, Guangxi Medical University Cancer Hospital, Nanning 530021, China; 4Department of Medicine, Harvard Medical School, Boston, MA 02215, USA

**Keywords:** RAD51, DNA damage, chemoresistance, immune dysregulation, colon cancer, breast cancer, solid tumor

## Abstract

**Simple Summary:**

Aims and objectives: Destabilization of genetic material can contribute to development and progression of cancer including development of drug resistance and treatment failure. We have previously shown that a protein RAD51 and its corresponding activity, which maintain stability of genetic material (DNA) in normal cells, are dysregulated in cancer. Purpose of this study was to investigate role of RAD51 and its potential as therapeutic target in cancer. Results: We demonstrate that increased level of RAD51 is associated with poor survival of esophageal, breast and colon cancer patients. Our data indicated that increased RAD51 contributes to growth as well as spontaneous and chemotherapy-induced damage and destabilization of genetic material in cancer cells. Moreover, the treatment of cancer cells with chemotherapeutic drug (cisplatin) led to activation of mechanisms which disrupt immune surveillance and maintenance of genetic material, whereas addition of RAD51 inhibitor prevented activation of such mechanisms. Conclusions: Drugs targeting RAD51 have potential to inhibit growth of cancer cells while preserving integrity of genetic material and immune system. When used in combination, these treatments can increase ability of chemotherapeutic drugs to kill cancer cells while minimizing their harmful impact on genetic material and immune system.

**Abstract:**

Background: In normal cells, homologous recombination (HR) is tightly regulated and plays an important role in the maintenance of genomic integrity and stability through precise repair of DNA damage. RAD51 is a recombinase that mediates homologous base pairing and strand exchange during DNA repair by HR. Our previous data in multiple myeloma and esophageal adenocarcinoma (EAC) show that dysregulated HR mediates genomic instability. Purpose of this study was to investigate role of HR in genomic instability, chemoresistance and immune dysregulation in solid tumors including colon and breast cancers. Methods: The GEO dataset were used to investigate correlation of RAD51 expression with patient survival and expression of various immune markers in EAC, breast and colorectal cancers. RAD51 was inhibited in cancer cell lines using shRNAs and a small molecule inhibitor. HR activity was evaluated using a plasmid-based assay, DNA breaks assessed by evaluating expression of γ-H2AX (a marker of DNA breaks) and p-RPA32 (a marker of DNA end resection) using Western blotting. Genomic instability was monitored by investigating micronuclei (a marker of genomic instability). Impact of RAD51 inhibitor and/or a DNA-damaging agent was assessed on viability and apoptosis in EAC, breast and colon cancer cell lines in vitro and in a subcutaneous tumor model of EAC. Impact of RAD51 inhibitor on expression profile was monitored by RNA sequencing. Results: Elevated RAD51 expression correlated with poor survival of EAC, breast and colon cancer patients. RAD51 knockdown in cancer cell lines inhibited DNA end resection and strand exchange activity (key steps in the initiation of HR) as well as spontaneous DNA breaks, whereas its overexpression increased DNA breaks and genomic instability. Treatment of EAC, colon and breast cancer cell lines with a small molecule inhibitor of RAD51 inhibited DNA breaking agent-induced DNA breaks and genomic instability. RAD51 inhibitor potentiated cytotoxicity of DNA breaking agent in all cancer cell types tested in vitro as well as in a subcutaneous model of EAC. Evaluation by RNA sequencing demonstrated that DNA repair and cell cycle related pathways were induced by DNA breaking agent whereas their induction either prevented or reversed by RAD51 inhibitor. In addition, immune-related pathways such as PD-1 and Interferon Signaling were also induced by DNA breaking agent whereas their induction prevented by RAD51 inhibitor. Consistent with these observations, elevated RAD51 expression also correlated with that of genes involved in inflammation and other immune surveillance. Conclusions: Elevated expression of RAD51 and associated HR activity is involved in spontaneous and DNA damaging agent-induced DNA breaks and genomic instability thus contributing to chemoresistance, immune dysregulation and poor prognosis in cancer. Therefore, inhibitors of RAD51 have great potential as therapeutic agents for EAC, colon, breast and probably other solid tumors.

## 1. Introduction

DNA in normal cells is exposed to a number of genotoxic agents which cause a variety of lesions [1,2] including abasic sites, missing bases, DNA strand breaks, DNA strand cross-links and deletions and insertions on daily basis. In normal healthy cells, diverse DNA repair mechanisms coordinate to accurately repair DNA damage in a timely and efficiently manner. The main repair mechanisms for double-stranded DNA breaks are homologous recombination (HR) and non-homologous end joining. HR, which uses homologous sequences from homologous chromosome or sister chromatid as template to repair the damage, is the most precise of all repair mechanisms [3]. In normal cells the process of HR is tightly regulated and utilized to accurately repair multiple types of DNA damage including double-stranded breaks, single-stranded gaps and strand crosslinks and, therefore, has a prominent role in the maintenance of genome stability and integrity [4]. Therefore, mutations and/or other defects leading to loss of HR function can lead to genomic instability and oncogenesis as seen for a subset of breast and ovarian cancers. Ovarian and breast cancer susceptibility (BRCA) genes which are mutated in a small subset (<1–7%) of these cancers [5], have been shown to coordinate with RAD51 to facilitate HR process [6].

However, dysregulation leading to overactivation of HR, a process which has ability to bring distant regions of DNA in close vicinity based on sequence homology, can also lead to genomic rearrangements and instability [7]. Our previous data in esophageal adenocarcinoma (EAC) and multiple myeloma demonstrated that HR activity is spontaneously elevated (dysregulated) and an important underlying cause of increased DNA damage [8] and genomic instability [9,10]. Our previous data also demonstrated that inhibition of HR, through chemical as well as transgenic inhibition of recombinase RAD51, reduces genomic instability as well as tumor growth in a subcutaneous model of EAC [11]. In addition to a key role in HR activity, RAD51 also contributes to other processes such as telomere stabilization [11]. Dysregulated HR and associated genomic instability have also been implicated in development of drug resistance in cancer [10]. Consistently, several key HR pathway intermediates have been shown to be overexpressed in different cancers [10,11,12,13,14]. Amplification and overexpression of RAD51C, a paralog of RAD51, has been demonstrated in breast cancer cell lines including MCF-7 and a subset of primary breast cancer samples [12]. Elevated RAD51 expression in lung cancer patients has been shown to correlate with their poor survival [15].

Overactive and dysregulated HR can lead to several aberrant forms of this repair process including homeologous recombination (with reduced precision because of less dependence on strict homology between sequences to be recombined), non-allelic HR (NAHR), and/or recombination events occurring at wrong place and time [16]. The dysregulated and aberrant HR events have been implicated in a variety of changes at DNA and chromosomal levels [17,18,19]. Ongoing genomic rearrangements and instability contribute to oncogenesis and cancer progression through their impact on multiple cellular processes. These genomic changes also impact immune surveillance at multiple levels. A crosstalk between DNA damage and immune response also exists [20,21]. Pattern recognition receptors which are key component of immune signaling not only detect nucleic acids of foreign origin but also recognize the damaged host DNA [16]. For example, recognition of unmethylated CpG motifs by a pattern recognition receptor TLR9 can induce inflammatory response, linking DNA maintenance with inflammation. On other hand, certain DNA damage response proteins can also function as pattern recognition receptors and can induce inflammatory response following detection of cytoplasmic DNA [22,23,24,25,26]. There is evidence that ATM, a major DNA damage response kinase, is also involved in immune signaling through regulation of transcription factors including NF-κB and iRFs. It has been demonstrated that activation of NF-κB activates HR-dependent repair of double strand DNA breaks [27].

In this manuscript we investigated recombinase RAD51 for its genomic impact as well as its association with clinical outcome in solid tumors including esophageal adenocarcinoma (EAC) and colon and breast cancers. We demonstrate that elevated RAD51 Expression correlates with poor survival of cancer patients in all three cancer types tested. Evaluation in EAC demonstrated that RAD51 expression correlates with several key genes involved in inflammatory and other immune signaling. Treatment of cancer cells with a DNA breaking agent led to upregulation of immune-related pathways such as PD-1 and Interferon Signaling whereas this upregulation was either prevented or reversed by RAD51 inhibitor. Inhibition of RAD51 reduced spontaneous and DNA breaking agent-induced DNA breaks and genomic instability whereas potentiated cytotoxicity of DNA breaking agents in all cancer cell types tested.

## 2. Materials and Methods

### 2.1. Bioinformatics Data Sources

Clinical data from three GEO datasets and one TCGA dataset were used. These include: 1-GSE13898: Containing 64 primary esophageal adenocarcinoma (EAC) and 28 surrounding normal fresh frozen tissue samples; 2-GSE33113: Primary tumor resections from 90 colon cancer patients and matching normal colon tissue from 6 of these patients; 3-GSE20685: Containing 327 breast cancer samples. 4: TCGA dataset: Esophageal carcinoma (ESCA) set included 185 samples. For survival analysis, we removed samples with “NA” in the survival parameter.

### 2.2. Cell Types

Normal human mammary epithelial (HMEC) cells were obtained from Lonza Bioscience (Walkersville, MD, USA), normal primary human esophageal epithelial (HEsEpiC) cells were from ScienCell Research Laboratories (Carlsbad, CA, USA) and normal human diploid fibroblasts (HDF) and cancer cell lines—colon (HCT116, HT29) and breast cancer (MCF7) cell lines purchased from American Type Culture Collection (Manassas, VA, USA). Esophageal adenocarcinoma (EAC) cell lines (FLO-1, OE19) were obtained from Sigma Aldrich Corporation (Saint Louis, MO, USA). Cells were cultured as reported previously [8,9,11,28].

### 2.3. Cell Viability and Apoptosis

Cell Titer-Glo Luminescent Viability Assay kit (Promega Corporation, Madison, WI, USA) was used to monitor cell viability. Control cells or those treated with RAD51 inhibitor and/or etoposide were washed and treated with fluorescein isothiocyanate-conjugated annexin V and propidium iodide. The percentage of cells undergoing apoptosis in each treatment cohort was analyzed by flow cytometry.

### 2.4. Modulation of RAD51 Expression

RAD51 was suppressed using shRNA (Clone ID:TRCN0000329686 purchased from MilliporeSigma, Burlington, MA, USA)-mediated knockdown or RAD51 inhibitor “RI-1” [CAS 415713-60-9; 3-chloro-1-(3,4-dichlorophenyl)-4-morpholino-1*H*-pyrrole-2,5-dione] purchased from Sigma-Aldrich, Milwaukee, WI, USA. The shRNA knockdown cell lines were selected in puromycin to obtain stably transduced cell lines. Anti-RAD51 antibody (CAS:sc-377467, Santa Cruz, Dallas, TX, USA) was used to confirm knockdown by Western blots.

### 2.5. Detection of DNA Breaks and End Resection with Different DNA Breaking Agents, Inhibitor or Gene Modulation

DNA breaking agents including camptothecin (CAS 7689-03-4, Sigma-Aldrich, St. Louis, MO, USA), 5-fluorouracil (5-FU, CAS 51-21-8—Calbiochem) or paclitaxel (PTX, CAS 33069-62-4—Calbiochem) were used. DNA breaks were monitored by measuring expression of γ-H2AX (CAS 80312S; Cell Signaling, Danvers, MA, USA), a marker of DNA breaks and RPA32 phosphorylated on ser4 and ser8 (pRPA32; CAS NBP1-23017; Novus Biologicals, Littleton, CO, USA), a marker of DNA end resection [29,30] by Western blotting. Western blot bands were quantitated using ImageJ 1.52a (National Institutes of Health, USA).

### 2.6. Strand Exchange Assay

Strand exchange, a distinct step in the initiation of homologous recombination (HR), was measured in the lysates of control and RAD51-knockdown cells by a fluorescence-based assay, as reported by Huang et al. [31].

### 2.7. Evaluation of RAD51 Inhibitor, Alone or in the Presence of a DNA Breaking Agent, In Vivo

CB-17 SCID mice were purchased from Charles River Laboratories (Wilmington, MA, USA) and housed following guidelines of the Institutional Animal Care and Use Committee (IACUC). The experimental protocol was approved by the IACUC and the Occupational Health and Safety Department of Dana Farber Cancer Institute, Boston, MA. OE19 cells were injected subcutaneously into SCID mice and following the appearance of tumors, mice treated with either DMSO, RAD51 inhibitor (RI-1; 25 mg/kg, 3 times/week for 2 weeks), cisplatin (CAS 15663-27-1—Calbiochem; 3 mg/kg once a week for 2 weeks) or in combination. Sizes of tumor were measured twice every week. Mice were euthanized when tumors became 2 cm^3^ in size or if any major problem or paralysis occurred.

### 2.8. Impact on Expression Profile

Esophageal adenocarcinoma (FLO-1) cells were treated with RAD51 inhibitor (RI-1; 20 μM), DNA breaking agent camptothecin (CPT; 100 nM) or combination of both (RI-1 + CPT) for 24 h and impact on gene expression monitored by RNASeq. Pathways which were induced or activated by CPT while their induction was either prevented or reversed by RI-1 were identified.

### 2.9. Investigating Genomic Instability

Genomic instability in live cell fraction was monitored by investigating micronuclei, a marker of genomic instability [32], by a flow cytometry-based assay using MicroFlow^®^ In Vitro Micronuclei Plus Kit (Cat#567,797 purchased from BD, Franklin Lakes, NJ, USA).

### 2.10. Statistics and Reproducibility

For validation of the impact of RAD51, both the gain and loss of function, RAD51 inhibitor, total of five cell lines (representing 3 different cancers), one normal cell strain, and the evaluation of different parameters of genome stability and growth were used. The impact of overexpression of RAD51 was demonstrated in normal human fibroblasts as well as in a breast cancer cell line. The impact of RAD51 inhibition (by shRNA and/or inhibitor) was demonstrated in four cell lines (representing 3 different cancers). Approaches to evaluate the genomic impact included the investigation of DNA end resection, spontaneous DNA breaks, strand exchange activity and micronuclei formation. For evaluation of the impact of RAD51 inhibitor in combination with chemotherapy, three different chemotherapeutic agents were used in vitro, and another used in a subcutaneous model of EAC using 5 mice per treatment group. All experiments were conducted in triplicate and information about the error bars provided in figure legends. Thus, using different cell types, DNA damaging agents and transgenic and chemical modulation of RAD51, we demonstrate that elevated RAD51 contributes to increased DNA breaks and its inhibitor reduces DNA breaks and genomic instability caused by different DNA damaging agents in multiple cancer cell types. RAD51 inhibitor also potentiated cytotoxicity of chemotherapy in vitro and in subcutaneous model of EAC.

## 3. Results

### 3.1. Elevated RAD51 Expression Correlates with Poor Survival

We investigated RAD51 expression in normal and cancer samples from clinical datasets of esophageal adenocarcinoma (EAC: GSE13898) and colon cancer (GSE33113) patients. Expression of RAD51 was significantly elevated in EAC and colon cancer samples relative to corresponding normal controls (*p* < 0.04) (Figure 1A). Elevated RAD51 expression also correlated with poor overall survival in two different datasets of EAC (GSE13898; cutoff: 6.237) and TCGA (cutoff: 363.63) as well as in colon (cutoff: 261) and breast cancer (cutoff: 6.31) (Figure 1B). These data indicate that elevated RAD51 expression is associated with poor clinical outcome in gastrointestinal and other tumors such as breast cancer.

### 3.2. Elevated RAD51 Expression Contributes DNA Breaks in Solid Tumors

Expression of RAD51 and phosphorylated-RPA32, a marker of DNA end resection a decisive step in the initiation of homologous recombination (HR) activity was evaluated in human cancer cell lines—esophageal adenocarcinoma (OE19, FLO-1), breast cancer (MCF7, MDA231, MDA435) and colon cancer (HCT116, HT29) relative to normal cell types (human diploid fibroblasts; human esophageal epithelial cells and human mammary epithelial cells). Relative to three normal cell types, both RAD51 and pRPA32 were overexpressed in gastrointestinal as well as breast cancer cell lines (Figure 2A). RAD51-knockdown in OE19 and HCT116 inhibited DNA breaks, as assessed from γ-H2AX expression, in both cell lines (Figure 2(BI)). Importantly, the loss of RAD51 expression in both cell lines was commensurate with the inhibition of γ-H2AX expression. RAD51 knockdown was also associated with inhibition of homologous strand exchange activity, a distinct step in the initiation of homologous recombination (HR) activity (Appendix A). Consistent with these observations, investigation in breast cancer cell line (MDA231) demonstrated that overexpression of RAD51 increases DNA breaks in cancer cells (Figure 2C).

We then investigated if inhibition of RAD51 could reduce or minimize genotoxic impact of different DNA breaking agents. Four cell lines (OE19, FLO-1, MCF7 and HCT116) representing three different cancers were treated with RAD51 inhibitor, alone or in the presence of a DNA breaking agent, either camptothecin (CPT; a known DNA breaking agent [33]), 5-fluorouracil (5-FU) or paclitaxel (PTX) as indicated in Figure 2C. Treatment of cancer cell lines (OE19, FLO-1, HCT116 and MCF7) with all these genotoxic agents caused an increase in DNA breaks, whereas addition of RAD51 inhibitor led to reduction in DNA damage caused by these agents (Figure 2D). Evaluation in CPT-treated OE19, HCT116 and MCF7 cell lines demonstrated that CPT caused an increase in homologous strand exchange activity, a distinct step in the initiation of HR activity which was significantly reduced by addition of RAD51 inhibitor (Appendix A). Overall, these data indicate that RAD51 and associated strand exchange/HR activity contribute to increased DNA breaks in gastrointestinal as well as other tumors.

### 3.3. RAD51 Inhibitor Significantly Reduces DNA Breaking Agent-Induced Genomic Instability in Cancer Cell Lines

To investigate the impact of RAD51 inhibitor on CPT-induced genomic instability, esophageal adenocarcinoma (OE19 and FLO-1), colon (HCT116) and breast cancer (MCF7) cell lines were treated with RAD51 inhibitor (RI-1), camptothecin (CPT; a DNA breaking agent) or combination of both for 48 h and the impact on micronuclei (a marker of genomic instability) evaluated using flow cytometry. CPT increased genomic instability in MCF7, HCT116, FLO-1 and OE19 cells by 1.5-, 1.8-, 3- and 7-fold, respectively whereas addition of RAD51 inhibitor significantly inhibited CPT-induced genomic instability in all cell types (Figure 3A–D).

Consistent with this, RAD51 overexpression in normal human fibroblast cells increased micronuclei (a marker of genomic instability) by ∼20-fold (Figure 3E). These data are consistent with our previous data demonstrating that inhibition of RAD51 in cancer cells [multiple myeloma (MM) and esophageal adenocarcinoma (EAC)], by transgenic and/or chemical modifications, inhibits HR and genomic instability [9,10,34] and etoposide-induced DNA damage [35]. Overall, these data show that RAD51 contributes to spontaneous and chemotherapy-induced genomic instability in cancer.

### 3.4. RAD51 Inhibitor Potentiates Cytotoxicity of DNA Breaking Agent in Cancer Cell Lines

Esophageal adenocarcinoma (OE19, FLO-1), colon (HCT116) and breast (MCF7) cancer cell lines were treated with RAD51 inhibitor (RI-1), camptothecin (CPT) or combination of both for 48 h and cell viability assessed as described in Methods. RAD51 inhibitor significantly potentiated cytotoxicity of CPT in all cancer cell types tested (Figure 4) and the impact of RAD51 inhibitor on CPT-induced cytotoxicity was mostly synergistic (Appendix A). Breast (MCF7), colon (HCT116) and esophageal adenocarcinoma (OE19) cell lines treated with RAD51 inhibitor, CPT or combination for 48 h were also evaluated for apoptosis using flow cytometry. In MCF7 and HCT116 cell lines, treatment with CPT caused apoptotic death in ∼38% cells whereas addition of R1 increased the apoptotic fraction to ∼58% (*p* < 0.05). Similarly, in OE19 cell line, the treatment with CPT caused apoptotic death in ∼41% cells whereas addition of R1 increased the apoptotic fraction to ∼55% (*p* < 0.05) (Figure 5).

### 3.5. RAD51 Inhibitor Increases Sensitivity of DNA Breaking Agent in EAC Cells In Vivo

The impact of RAD51 inhibitor on efficacy of DNA breaking agent was also investigated in a subcutaneous tumor model of EAC. OE19 cells were injected subcutaneously into SCID mice and following the appearance of tumors, mice treated with either DMSO, RAD51 inhibitor (RI-1; 25 mg/kg, 3 times/week for 2 weeks), cisplatin (3 mg/kg, once a week for 2 weeks) or in combination. Line plots showing tumor growth following different treatments are shown in Figure 6. Relative to control (vehicle treatment), tumor sizes in mice treated with RAD51 inhibitor, cisplatin and combination of RAD51 inhibitor and cisplatin were reduced by 30%, 59% and 71%, respectively (*p* < 0.04) (Figure 6). Relative to tumors in mice treated with RAD51 inhibitor or cisplatin, the mice exposed to combination treatment had significantly smaller tumors (*p* < 0.03).

### 3.6. Association of RAD51 with Immune Dysregulation

EAC (FLO-1) cells were treated with RAD51 inhibitor, DNA breaking agent (camptothecin; CPT) or combination for 24 h and impact on gene expression monitored by RNASeq. Pathways which were induced or activated by CPT (green bars) while their induction was either prevented or reversed by RAD51 inhibitor (red bars) are shown in Figure 7. Several DNA repair and cell cycle related pathways including Homology Directed Repair, DNA Strand Elongation, Mitotic, Head and Neck Cancer and TP53 and TP63 Targets and Cell Cycle were induced by CPT whereas their induction either prevented or reversed by RAD51 inhibitor.

Interestingly, certain immune-related pathways such as PD-1 and Interferon Signaling were also induced by CPT whereas their induction either prevented or reversed by RAD51 inhibitor. Based on these observations, we investigated if RAD51 expression correlates with immune components involved in inflammation and/or immune surveillance in esophageal adenocarcinoma (dataset; GSE13898). Elevated RAD51 expression correlated with increased expression of various immune components including interferon γ, IL6, IL1β, IL2, TNFα, PD-1, PD-L1, OX40 and OX40L (Table 1; line plots of correlation shown in Appendix A). These data suggest that elevated RAD51 expression is commensurate with immune dysregulation.

Elevated RAD51/HR-associated genomic instability can impact immune function by activating cGAS-STING pathway. We, therefore, investigated if RAD51 overexpression impacts cGAS expression. RAD51 overexpression in normal human diploid fibroblast (HDF) cells caused a marked increase in genomic instability (as assessed from increase in micronuclei) (Figure 3E). However, no increase in cGAS expression was observed. In fact, cGAS expression was reduced following overexpression of RAD51 in normal human diploid fibroblasts (Appendix A). Consistently, the overexpression of RAD51 in breast cancer cells led to increase in DNA breaks and loss of cGAS expression (Figure 2C). Overexpression of RAD51 in EAC (FLO-1) cells also led to loss of cGAS expression (Appendix A).

### 3.7. Impact of Elevated RAD51 on Cell Cycle

We overexpressed RAD51 in normal (normal human diploid fibroblasts; HDF), EAC (FLO-1) and breast cancer (MDA-MB-231) cells and investigated the impact on cell cycle using flow cytometry. Although RAD51 overexpression in normal HDF and breast cancer cells markedly disrupted genomic integrity (Figure 3E and Appendix A), no impact on cell cycle was observed in these cell types (Appendix A). However, in EAC (FLO-1) cells the overexpression of RAD51 was associated with >2-fold increase in the fraction of cells in S phase, whereas ~25% reduction in the cells in G1.

## 4. Discussion

Hallmarks of cancer include persistent proliferative signals, inactivation or loss of growth resistance genes, prevention of cell death/apoptosis, acquisition of indefinite lifespan, angiogenesis and acquisition of ability to invade and metastasize. Genetic instability enables cells to acquire genomic changes required for all these processes. Genomic instability and ongoing genomic changes can also enable affected cells to overcome immune surveillance and develop resistance to treatment. Some of the mechanisms driving genomic instability also drive cell cycle progression and growth of cancer cells. Thus, genomic instability is a potential target to inhibit tumor development, growth and chemotherapy resistance. Data from our laboratory demonstrate that cancer cell lines have increased spontaneous DNA damage (https://www.biorxiv.org/content/10.1101/2022.04.20.488830v1.supplementary-material, accessed on 22 April 2022). Our previous data in EAC and multiple myeloma model systems have demonstrated that elevated homologous recombination (HR) activity contributes to increased DNA damage [8] and genomic instability [9,10]. Moreover, inhibition of HR, through chemical as well as transgenic inhibition of recombinase RAD51, reduces genomic instability as well as tumor growth in a subcutaneous model of EAC [11].

In normal cells, cell cycle checkpoints and other regulatory mechanisms ensure that HR is precise, well-regulated and occurs preferably in G2 phase of cell cycle [16]. However, this is not true for spontaneously elevated (dysregulated) RAD51/HR in cancer situation. For example, p53 (a key cell cycle regulator), associates with RAD51 to keep it in an inactive state whereas its sequestration by oncogene in immortal cells [36] and/or mutation/s in this gene, disrupt p53-RAD51 association thus leading to increased/dysregulated RAD51/HR. On the other hand, overactive HR can also override and dysregulate cell cycle. For example, RAD54B, a RAD54 homologue shown to interact with RAD51 [37] and involved in HR process [28], is overexpressed in several cancers including EAC and involved in p53 degradation and cell cycle progression in cancer [38]. Thus dysregulated, unscheduled and unnecessary recombination events and loss of checkpoint controls (whether mediated by drivers of genomic instability such as RAD54B and/or other mechanisms) are attributed to genomic instability in cancer.

In this study, we further investigated consequences of elevated RAD51 and HR in EAC as well as other cancers. Suppression of RAD51 in cancer cell lines inhibited HR-related strand exchange activity and spontaneous as well as DNA breaking agent-induced DNA breaks and genomic instability in live cell fraction. Consistently, RAD51 overexpression in breast cancer cells increased DNA breaks. Moreover, RAD51 overexpression also increased genomic instability in normal human fibroblast cells by several fold. Importantly, the genomic instability caused by three different drugs (camptothecin [33], a topoisomerase inhibitor; 5-fluorouracil, a thymidylate synthase inhibitor; and paclitaxel, an antimicrotubule agent), which induce DNA breaks by different mechanisms, was minimized by addition of RAD51 inhibitor in cell lines representing three different cancers (EAC, colon and breast). These data suggest that RAD51-dependent HR is a common mechanism of genomic instability following increased DNA breaks, irrespective of the source of these breaks. These observations also serve to corroborate our previous observations in EAC in which investigation of mutational signatures identified HR as a common mechanism of genomic instability following overexpression of three different genes in normal human cells [28]. Our published and unpublished data also show that besides RAD51, the suppression of other HR proteins APE1, RAD54B, FEN1 and EXO1 also inhibit DNA damage and genomic instability [28,34,39] thus further confirming the role of HR in this process.

RAD51 inhibitor also significantly potantiated cytotoxicity of DNA breaking agent in all cancer cell types tested in vitro. RAD51 inhibitor also inhibited tumor growth in subcutaneous mouse model of EAC without any toxicity to mice as well as significantly potentiated cytotoxicity of DNA breaking agent in vivo. This is consistent with the role of RAD51 in dysregulation of HR and genome stability and supports our previous in vitro data in EAC [28]. Although dysregulated (elevated) RAD51 could impact growth of cancer cells through possible impact on cell cycle, no changes in cell cycle were detected following overexpression of RAD51 in normal (HDF) or a breast cancer cell line tested (Appendix A). However, in EAC (FLO-1) cells the overexpression of RAD51 was associated with substantial increase in the fraction of cells in S phase and reduction in the percentage of cells in G1. These data indicate that although dysregulated RAD51 may impact cell cycle in certain cell types (such as FLO-1), it does not necessarily impact genome stability and growth of cancer cells through alterations in cell cycle. It may impact these processes by other mechanism/s. These may include the role of RAD51/HR in genomic rearrangements, telomere stabilization and their ability to conceal DNA breaks (from apoptosis machinery) either through their accurate repair or by engaging them in genomic rearrangements etc.

Evaluation by RNA sequencing demonstrated that DNA repair and cell cycle related pathways were induced by DNA breaking agent camptothecin (CPT), whereas their induction either prevented or reversed by RAD51 inhibitor. CPT is a topoisomerase I inhibitor and it has been proposed that replication-dependent DNA breaks induced by this drug are primarily fixed by HR [33]. Consistent with the known function of CPT, the treatment of cancer cells with this drug led to upregulation of several DNA repair and cell cycle related pathways including Homology Directed Repair, DNA Strand Elongation, Head and Neck Cancer and TP53 and TP63 Targets and Cell Cycle. Moreover, addition of RAD51 inhibitor either prevented or minimized the induction of these pathways by CPT. These data suggest that pathways upregulated following exposure to CPT are a consequence of increased HR and, therefore, their induction is prevented when CPT is combined with RAD51 inhibitor.

Interestingly, certain immune-related pathways such as PD-1 and Interferon Signaling were also induced by CPT whereas their induction either prevented or reversed by RAD51 inhibitor. Consistently, the investigation in EAC dataset (GSE13898) demonstrated that elevated RAD51 expression correlates with increased expression of various genes involved in immune signaling. These include inflammatory cytokines (interferon γ, IL6, IL1β, IL2, TNFα), checkpoint inhibitors (PD-1, PD-L1, TIGIT, TIM3) as well as some costimulatory molecules (OX40 and OX40L). Multiple mechanisms exist which enable tumor cells escape immune surveillance. One of such mechanisms depends on checkpoint inhibitory receptors (such as PD-1, TIGIT) which when upregulated on T cells, limit their survival and function. Elevated RAD51/HR-associated genomic instability can impact immune function by activating cGAS-STING pathway. However, we observed that RAD51 overexpression in normal as well as cancer cells was associated with increased DNA breaks and genomic instability but reduced cGAS expression (Figure 3E and Appendix A). Since cGAS-STING pathway is the cytosolic double-stranded DNA sensor, our data suggest that overexpression of RAD51 probably inhibited or reduced the release of DNA fragments (produced from rupture of micronuclei or other genomic rearrangements) into cytoplasm. It is possible that increased HR activity following RAD51 overexpression engaged these DNA/chromosomal fragments into aberrant recombination and other genomic events such as chromothripsis, thus reducing their release into cytoplasm. Therefore, elevated RAD51/HR-associated impact on immune system may involve mechanisms other than cGAS/STING. There is substantial evidence of a functional link between genomic instability and inflammation [16]. Genomic instability can lead to inflammation whereas inflammation can increase genomic instability. It is, therefore, possible that RAD51/HR-associated genomic instability contributes to dysregulation of immune surveillance.

Several key HR pathway intermediates have been shown to be overexpressed in different cancers [10,11,12,13,14]. Our previous investigations demonstrated elevated RAD51 expression in multiple myeloma and EAC cell lines and patient samples. In this study, we evaluated both RAD51 and pRPA32 (a marker of DNA end resection, a distinct step involved in the initiation of HR activity) and demonstrate that the levels of both these proteins are high in EAC (OE19, FLO-1), colon cancer (HCT116, HT29) and breast cancer (MCF7) and cell lines. Amplification and overexpression of RAD51C (a paralog of RAD51) has also been demonstrated in breast cancer cell lines including MCF-7 and a subset of primary breast cancer samples [12]. In addition to well-known role of RAD51 in HR process, there is evidence that it also contributes to other processes such as telomere stabilization [11]. Our unpublished data in multiple myeloma also suggest a role of RAD51 in DNA replication. Thus, elevated RAD51 expression can help cancer cell survival and evolution by acting at multiple levels. Consistent with this, we here demonstrate that elevated RAD51 expression correlates with poor survival in EAC, colon and breast cancer patients. Elevated RAD51 expression in lung cancer patients has also been shown to correlate with their poor survival [15].

Elevated/dysregulated RAD51/HR can lead to several aberrant forms of repair including homeologous recombination (with reduced precision because of less dependence on strict homology between sequences to be recombined), non-allelic HR (NAHR), and/or recombination events occurring at wrong place and time [16]. Such aberrant HR events have been implicated in a variety of changes at DNA and chromosomal levels [17,18,19]. However, mutations and/or other defects leading to loss of HR function can also lead to genomic instability and oncogenesis as seen for a subset of breast and ovarian cancers. Ovarian and breast cancer susceptibility (BRCA) genes which have been shown to coordinate with RAD51 to facilitate HR process [6], are mutated in a small subset (<1–7%) of these cancers [5] thus contributing to genomic instability because of the loss of HR. Ongoing genomic rearrangements and instability contribute to oncogenesis and cancer progression through their impact on multiple cellular processes including immune dysregulation.

An important question is that how RAD51/HR inhibition stabilizes genome while increasing cytotoxicity of chemotherapy. Cancers such as EAC and multiple myeloma are associated with increased DNA damage, HR activity and genomic instability. Elevated RAD51 and HR in these cells are not only implicated in genomic rearrangements and instability but also contribute to their growth and survival by various mechanisms. These may include their role in telomere stabilization, telomere elongation and their ability to conceal DNA breaks (from apoptosis machinery) either through their accurate repair or by engaging them in genomic rearrangements. Treatment with a chemotherapeutic agent further increases DNA damage in cancer cells. The cells which acquire extensive DNA breaks are killed, whereas those with a small or (manageable) increase in DNA damage, survive the treatment. The increased DNA breaks in surviving cells lead to further increase in HR activity which helps in their survival by eliminating/concealing DNA breaks. However, elevated HR also contributes to unnecessary/aberrant recombination events leading to increase in genomic instability in these surviving cells. When chemotherapy is combined with RAD51 inhibitor, the HR activity in treated cells is reduced. This leads to: (1) Increase in the number of cells with extensive DNA damage thus increasing the fraction of cells which are killed; (2) Reduced HR activity and associated genomic instability in the surviving cells.

We, therefore, conclude that elevated expression of RAD51 and associated HR activity is involved in spontaneous and DNA damaging agent-induced DNA breaks and genomic instability thus contributing to chemoresistance, immune dysregulation and poor prognosis in cancer. Inhibitors of RAD51 have great potential as therapeutic agents for EAC, colon, breast and probably other solid tumors.

## Figures and Tables

**Figure 1 cancers-14-05697-f001:**
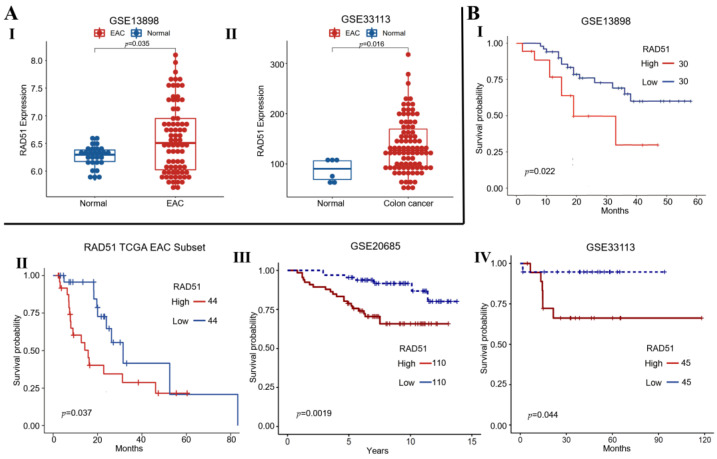
Elevated RAD51 expression correlates with poor survival of cancer patients. (**A**) RAD51 expression in normal vs. cancer samples from esophageal adenocarcinoma (EAC: GSE13898) (I) and colon cancer (GSE33113) (II) patients; (**B**) Survival curves show that elevated RAD51 expression correlates with poor overall survival in EAC—GSE13898 (I), EAC—TCGA (II), breast—GSE20685 (III), and colon cancer—GSE33113 (IV) patients.

**Figure 2 cancers-14-05697-f002:**
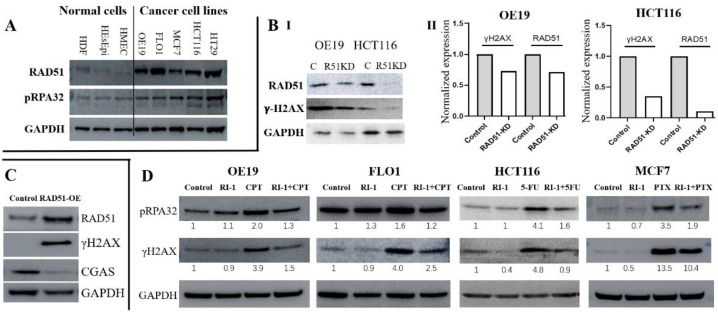
Elevated RAD51 contributes to spontaneous and induced DNA breaks in multiple solid tumors. (**A**) Western blot showing expression of RAD51 and phosphorylated-RPA32 (a marker of DNA end resection, a decisive step in HR) in normal cell types (HDF, human diploid fibroblasts; HEsEpi, human esophageal epithelial cells; HMEC, human mammary epithelial cells) and human cancer cell lines—esophageal adenocarcinoma (OE19, FLO-1), breast cancer (MCF7) and colon cancer (HCT116, HT29). (**B**) Rad51 was suppressed in cancer cell lines (OE19, HCT116) using shRNA and cells selected in puromycin. (I) Western blot shows the expression of RAD51 and γ-H2AX (a marker of DNA breaks) in control (C) and RAD51-knockdown (R51KD) cells; (II) Bar graphs show expression of γ-H2AX and RAD51 normalized to GAPDH; (**C**) Western blot shows the expression of RAD51, γ-H2AX (a marker of DNA breaks) and CGAS in control and RAD51-overexpressing (RAD51-OE) breast cancer (MDA231) cells; (**D**) Western blot showing expression of phosphorylated-RPA32 and γ-H2AX in different cancer cell lines treated with DMSO (control), RAD51 inhibitor (RI-1; 20 μM) alone or in the presence of DNA breaking agents—camptothecin (CPT; 100 nM), paclitaxel (PTX; 50 nM) or 5-fluorouracil (5FU; 5 μM) for 48 h.

**Figure 3 cancers-14-05697-f003:**
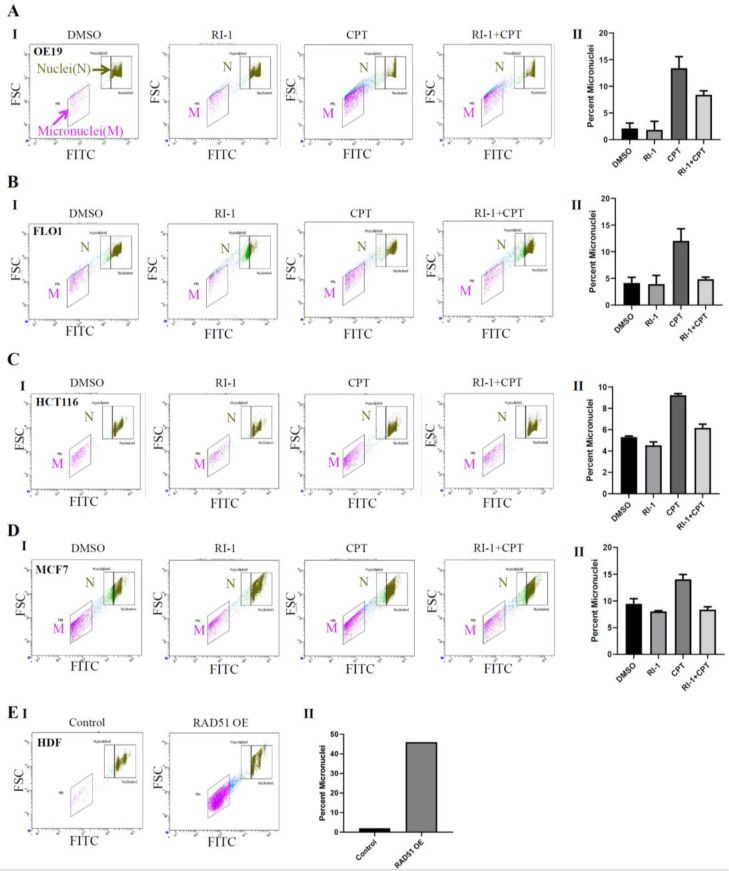
RAD51 inhibitor significantly reduces camptothecin-induced genomic instability in cancer cell lines. (**A**–**D**) Cancer cell lines—esophageal adenocarcinoma (OE19 and FLO-1) (**A**,**B**), colon (HCT116) (**C**) and breast cancer (MCF7) (**D**) were treated with RAD51 inhibitor (RI-1; 20 μM), camptothecin (CPT; 100 nM) or combination of both (RI-1 + CPT) for 48 h and evaluated for impact on micronuclei (a marker of genomic instability). Flow cytometry images of micronuclei (I) and bar graphs showing percentage of micronuclei (II) are shown; errors bars represent SDs of experiments conducted in triplicate; error bars indicate SDs of experiment conducted in triplicate. (**E**) RAD51 was overexpressed in normal human fibroblasts and cells evaluated for impact on micronuclei (a marker of genomic instability). Flow cytometry images of micronuclei (I) and bar graph showing percentage of micronuclei (II) are shown.

**Figure 4 cancers-14-05697-f004:**
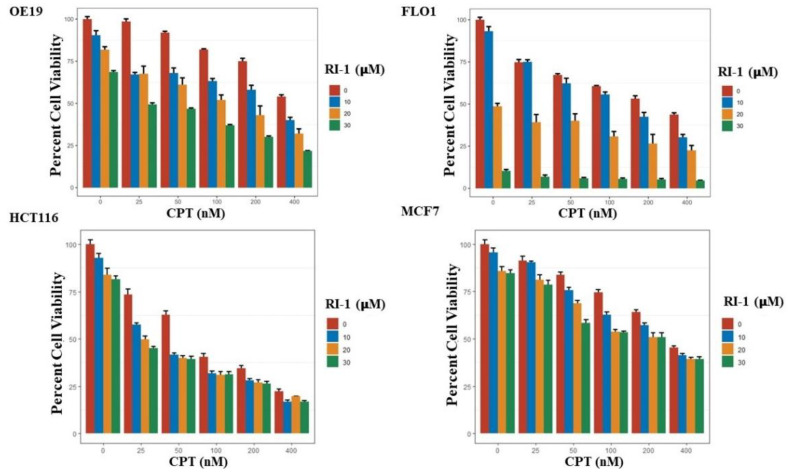
RAD51 inhibitor potentiates cytotoxicity of camptothecin in cancer cell lines. Esophageal adenocarcinoma (OE19, FLO-1), colon (HCT116) and breast (MCF7) cancer cell lines were treated with RAD51 inhibitor (RI-1), camptothecin (CPT) or combination of both (RI-1 + CPT) at different concentrations for 48 h and evaluated for cell viability as described in Methods. Errors bars represent SDs of triplicate assays.

**Figure 5 cancers-14-05697-f005:**
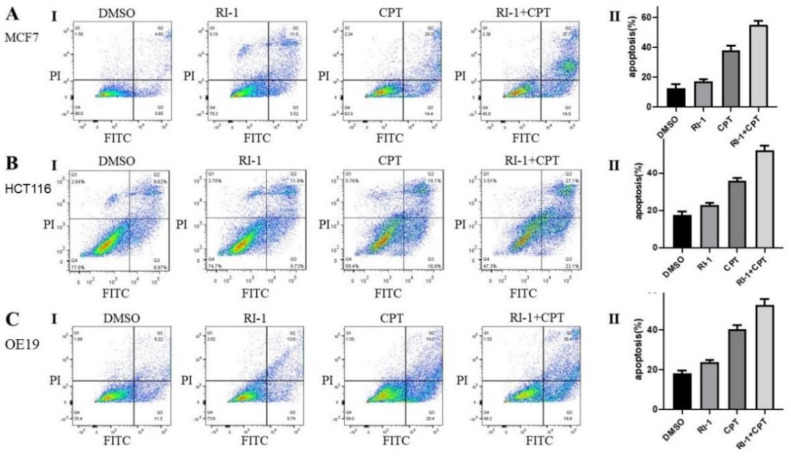
RAD51 inhibitor increases camptothecin-induced apoptosis in cancer cell lines. (**A**–**C**) Breast cancer (MCF7), colon cancer (HCT116) and esophageal adenocarcinoma (OE19) cell lines were treated with RAD51 inhibitor (RI-1; 20 μM), camptothecin (CPT; 100 nM) or combination of both (RI-1 + CPT) for 48 h and impact on apoptosis assessed using flow cytometry. Panels: (I) Flow cytometry images of apoptosis data; (II) Bar graphs showing percentage of apoptotic cells; error bars represent SDs of triplicate assays.

**Figure 6 cancers-14-05697-f006:**
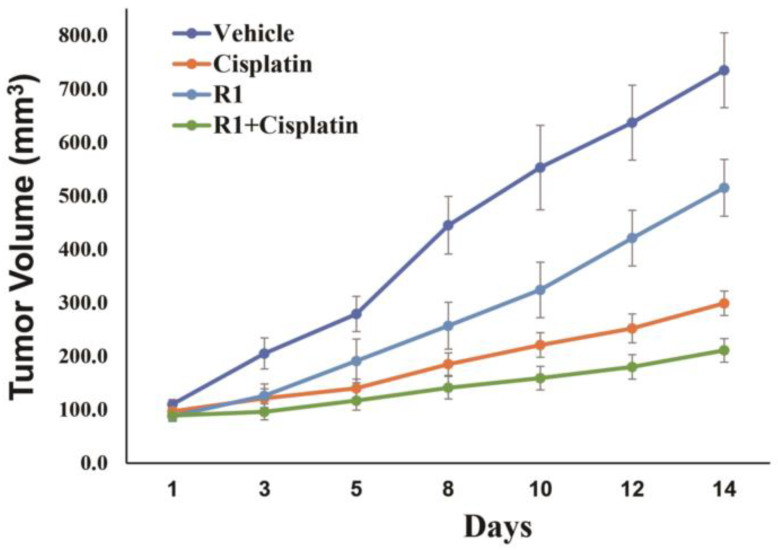
RAD51 inhibitor increases cisplatin sensitivity in EAC cells in vivo. OE19 cells were injected subcutaneously into SCID mice (5 mice per group) and following the appearance of tumors, mice treated with either DMSO, RAD51 inhibitor (RI-1; 25 mg/kg, 3 times/week for 2 weeks), cisplatin (3 mg/kg, once a week for 2 weeks) or in combination. Line plots showing tumor growth following different treatments are shown.

**Figure 7 cancers-14-05697-f007:**
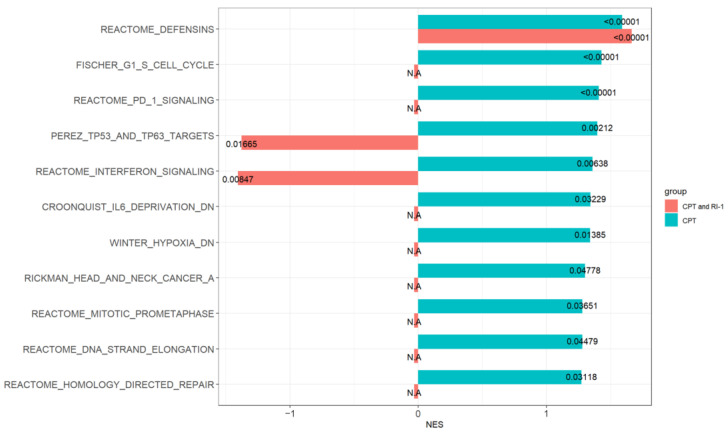
Impact of RAD51 inhibitor on camptothecin-induced changes in gene expression in cancer cells. Esophageal adenocarcinoma (FLO-1) cells were treated with RAD51 inhibitor (RI-1; 20 μM), camptothecin (CPT; 100 nM) or combination of both (RI-1 + CPT) for 24 h and impact on gene expression monitored by RNASeq. Green bars show pathways which were upregulated by CPT and red bars show the impact of RI-1 on their upregulation.

**Table 1 cancers-14-05697-t001:** Elevated RAD51 expression is commensurate with immune dysregulation. RAD51 expression was evaluated for correlation with various immune components involved in inflammation and cellular immunity in Esophageal adenocarcinoma (dataset; GSE13898). Elevated RAD51 expression correlated with increased expression of various immune components.

Marker Type	Marker	Correlation Coefficient	*p* Value
Inflammatory	GMCSF	0.32	0.0004
Inflammatory	IFNγ	0.62	4.4 × 10^−14^
Inflammatory	IL6	0.38	2.6 × 10^−5^
Inflammatory	IL1β	0.4	7.1 × 10^−6^
Inflammatory	IL2	0.6	8.6 × 10^−13^
Inflammatory	TNFα	0.37	4.2 × 10^−5^
Inflammatory	IL13	0.52	1.9 × 10^−9^
T cell	OX40	0.6	7.9 × 10^−13^
T cell	OX40L	0.63	1.5 × 10^−14^
T cell	TIGIT	0.55	1.8 × 10^−10^
T cell	PD-L1	0.55	1.7 × 10^−10^
T cell	PD1	0.71	2.2 × 10^−16^
T cell	TIM3	0.63	2.1 × 10^−14^
B lymphocyte	HLADRA	0.62	9.2 × 10^−14^
T, NK and B cells	CD27	0.51	3.1 × 10^−9^

## Data Availability

Appendix A is provided with the manuscript. RNASeq data is being deposited on Harvard dataverse.

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
