# Peer review of "RAD51 Is Implicated in DNA Damage, Chemoresistance and Immune Dysregulation in Solid Tumors"

_cancers, 2022, doi:10.3390/cancers14225697_

Round 1

Reviewer 1 Report

The manuscript of Liao et al. follows authors’ research of negative impact of hyperactive RAD51 and homologous recombination on genome stability in cancer. Initially, authors correlate the RAD51 expression levels with patients survival. Afterwards, various methods are used to demonstrate the decreased DNA damage when RAD51 is depleted/inhibited. Finally, authors show that RAD51 inhibition downregulate immune response triggered by DNA damage. Overall, this could be a good work but I have to admit, I am not satisfied with the proofs and the conclusions made from them, Materials and methods section was disappointing.

Major points:

The whole manuscript is based on that elevated levels of RAD51 lead to DNA damage and poor prognosis. Authors in this study use cell lines with increased RAD51 levels. It would be good to validate these results with mild overexpression of RAD51 in some control cells (noncancerous/cancerous) and see if it is also associated with elevated DNA damage.

RAD51 is quite important protein its depletion has serious consequences, as altering the cell cycle progression. Accumulation in G1 or G2 phase could lower the DNA damage markers because most of the endogenous damage occurs during S-phase. I suggest to quantify the cell cycle after RAD51 depletion and inhibition. This is a major point for the review.

If the negative effects of elevated RAD51 are truly because of hyperactive HR, maybe the effect of RAD51 depletion/inhibition could be mimicked by depletion/inhibition of another HR protein, which will not have such effect on cell cycle. Optionally, perform epistatic analysis between RAD51 and other HR protein.

In this manuscript authors show that RAD51 inhibition/depletion rather decrease DNA damage with or without DNA damaging compounds. That is interesting, I would expect the opposite results, as it is shown in cited publication number 8, where both mild and severe RAD51 depletion elevated markers of DNA damage. Decreased levels of micronuclei could be explained by prevention of fork reversal and thus shifting the stalled replication fork into restart. Maybe this could be discussed in the discussion section. Cell cycle alteration could also explain lowered numbers of micronuclei as they are created after mitosis.

The immune dysregulation could be an interesting point, but I think it needs some additional work. Authors should test whether the cGAS-STING pathway is activated in the presence of elevated RAD51 levels and suppressed upon inhibition/depletion. cGAS-STING pathway is activated by extranuclear DNA, so should be activated in presence of micronuclei. This pathway could be responsible for the immune response. I believe this experiment could increase the impact of the publication.

Materials and methods should be more descriptive. It is not necessary to describe the procedure but at least used methodology and basic facts should be stated. Sequence of shRNA and basic information about antibodies (company, type, catalogue number) should also listed. How quantification of Western blots was made? How many replicates were done? This is a major point for the review.

Method of micronuclei evaluation refers to previous publication, however, in cited publication, there is again reference to another publication only to find out that it was performed by commercial kit. The reader must open two additional publications to see the name of the kit which could be stated in MM of this manuscript! I feel that the way it is currently in the manuscript is only to get more citations, which I do not consider very ethical.

Minor points:

Page 4, line 155 – Typo: prreviously

Figure 2 – how long were the cells depleted? How long were the cells treated with RI-1? This is crucial information. The length of treatment/depletion will affect cell cycle. In Fig 3, there is the length of treatment. But is it the same as in fig2?

Figure 2 – how long was the treatment with CPT or 5-FU?

Figure2 BII – Normalized expression is not applicable for gH2AX. The same, p.13, line 380: pRPA32 is not overexpressed but phosphorylated

Figure2 BII – How the quantification was made? It should be at least from three experiments and it should be shown with error bars.

Author Response

POINT BY POINT RESPONSE TO REVIEWER 1 ATTACHED

Reviewer 2 Report

In this manuscript Liao et al. investigate the impact of RAD51 on genomic instability, chemo-sensitization and immune response in solid tumors: esophageal carcinoma (EAC), colon and breast cancer.

The authors show that overexpression of RAD51 is observed in EAC and colon cancer and correlates with poor prognosis in EAC, colon and breast cancer. Then the authors show that RAD51 overexpression is associated with increased pRPA in tumor cell lines. Reversely, inhibition or silencing of RAD51 induced a decrease in DSB formation (or signaling as it is monitored by gH2AX) and RPA phosphorylation. The use of RI-1, a RAD51 inhibitor decreases the induction of micronuclei and increases CPT induced apoptosis observed after CPT treatment in several cell lines and the size of tumors in EAC injected mice. Therefore, according to the authors, the decrease of DSB occurrence upon RAD51 inhibition is associated with a decreased cell survival. Finally, the authors report that inhibition of RAD51 dysregulate immune response in CPT treated cancer cells.

The questions asked in this manuscript are interesting but the replies are surprising and more importantly, incomplete.

Usually, inhibition of DSB repair leads to accumulation of unrepaired DSBs and ultimately cell death. Then a little bit of mechanistic or at least discussion and hypothesis on i) how RAD51 inhibition leads to decreased DSB and genomic instability, and ii) how decreased DSBs and genomic instability lead to more cell death would have been necessary. The authors do not comment except with one sentence in the discussion that is rather vague (“some of the mechanisms driving genomic instability also drive cell cycle progression and growth of tumor cells”) and they do not investigate the mentioned mechanisms.

It is expected that inhibition of RAD51 increases the toxicity of CPT as CPT induced lesions are mostly repaired by HR, but RAD51 inhibition is then supposed to lead to unrepaired DSBs. In fact, the decrease in the markers of DNA double strand breaks and resection upon RAD51 inhibition are rather surprising and that’s why they are interesting. But it is frustrating that the data and the information given in the manuscript do not sustain the conclusions solidly enough.

First of all there is a crucial lack of technical specifications making the results difficult to accept as we don’t know how (and how many times) they were obtained:

-       There is no mention of the number of independent experiments (which is not the same as “triplicates” as mentioned in figure 4) for each set of data, neither of the number of animals in each group of Figure 6.

-       The sequence/catalog number of the shRNA is missing.  Is it stably or transitory introduced in cells? If stable, what is the antibiotic for selection, etc…

-       HR reporter: the reference Huang et al. is only in vitro strand exchange and D-loop assays with purified proteins. This is not what we see here: can the authors describe precisely how they do the assay and add a scheme in the supplemental figure?

Besides, HR is not only strand exchange and has many subsequent steps in cells. So please be precise and mention strand exchange assays instead of HR on the figures.

-       No catalog number, provider, dilutions are given for antibodies. Same comment for inhibitors and drugs.

-       The method that was used for micronuclei is a commercial kit and not a method developed by the authors as the mentioned reference let the reader think. So please mention the provider and catalog number, instead of refering to a previous paper.

-       The number of samples in each GSE dataset is certainly available online but it would be useful to include it in the Mat/Meth section.

Figure 1B: please mention the cut-off and number of patients in each category High/Low (this is important to evaluate the relevance of the threshold) and homogenize the titles and legend.

Title 3.2: “Elevated RAD51 expression contributes to DNA breaks in solid tumors”.

This title does not correspond to the data shown. All experiments are performed in cell lines which is certainly different from tumors, so please change for “in cancer cell lines”. Second, most of the figure reports the impact of RAD51 inhibition by shRNA or inhibitor and not overexpression. The only part of the figure showing the impact of RAD51 overexpression (Figure 2A) does not show a proper marker of DSBs (like g-H2AX for example) but pRPA which is more a marker of resection.

Figure 2: The expression of RAD51 as well as the induction of DSBs (spontaneous or induced with the DNA damaging agents used here) are cell cycle regulated. On the other side, DNA damaging agents and RAD51 inhibition do influence the cell cycle distribution.

Then, it is necessary to see the cell cycle distribution and proliferation rate of the cell lines used in Figure 2A and the cell cycle distribution of cell lines treated with RAD51 shRNA or inhibitor with or w/o DNA damaging agent (Figure 2B). It could also be useful to show the expression of RAD51 in Figure 2C to show whether DNA damaging agents have an impact on RAD51 expression (via cell cycle modifications).

Figure 2A: pRPA32 is indeed a marker of resection, a decisive step of HR but upstream of RAD51 in the process. Increased pRPA could indicate that there is an increase in HR (starting at the resection step), but also that there is an increase occurrence of DSBs in these cells (that could be due to a higher proliferation rate for example).

Can the authors show the g-H2AX signal in cell lines of Figure 2A?

Figure 2B: Can teh authors show the pRPA32 signal in Figure 2B I?

Why are the DNA damaging agents different with different cell lines: either the cell line changes or the drug but not both. The authors should perform the experiments with CPT in all cell lines (HCT116 and MCF7 are then missing) and then use additional experiments with 5-FU and PTX as a confirmation that results are the same with another DNA damaging agent.

Figure 2B, 3, 4, 5: Was the purpose of these experiments to show that RAD51 inhibition is correlated to both a decrease in DSB/resection and a counterintuitive decrease in survival only when RAD51 is overexpressed at the beginning? If so, the authors should have included as for comparison, a panel of normal cells (like the ones in Figure 2A) in the experiments with RAD51 inhibitor and/or shRNA.

Figure 3: here again the impact of treatments on cell cycle will be informative because micronuclei implicate that cells are capable of undergoing mitosis, which is not necessarily the case upon RAD51 inhibition or CPT treatment.

Figure 4: Based on the data, I do not agree with the authors’ statement that “RAD51 inhibition increases cytotoxicity of CPT”, which suggest that there is a potentiation of CPT by RI-1.

In fact, my interpretation of the data is that what we see here is the simple addition of the toxicities of both agents RI-1 and CPT and in spite of the fancy -and unexplained- graphs shown in Supp. Figure 3, there is no synergy to me. For example, with OE19 cells; RI-1 at 30µM induces a decrease in cell viability of about 35%. CPT at 100µM decreases the viability of cells by 20%. The combination of the two induces a toxicity of 65%. Is it really different from the simple addition 35+20%?  

For the same concentrations in HCT116, the values are 20% toxicity with RI-1, 60% wit CPT and 70% with the combination. Is 20+60% really different from 70%? In the case it is significant, it does not support the authors’ conclusion.

Here, the reproduction of data in independent experiments (and not triplicates as mentioned in the legend) and statistical analysis are absolutely required.

The legends of figures 4 and 5 were inverted.

Figure 5: In contrast with Figure 4 and, as here RI-1 does not really increase apoptosis by itself, the authors can indeed conclude that the inhibition of RAD51 potentiates the toxicity of CPT. Nevertheless I have the same comment as for previous figure : independent experiments are required -and not triplicates- and statistical analysis.

There is a weird block (Figure 6.tif Type: TIF file, etc…) inserted in the middle of the figure.

Figure 6: Here again difficult to conclude when we have no indication on the number of animals in each group.

Figure 7: Why is the sample RI-1 alone not shown (although it is mentioned in the title of the figure –“impact of RAD51 inhibition on spontaneous…changes in gene expression”)? It is of course interesting to show that RI-1 abolishes the impact of CPT on the expression of these genes but it would also be interesting and informative to know if it corresponds to the effect of the inhibitor alone, whether it is combined to DNA damaging agents or not.

Introduction and discussion do not mention papers already existing on the correlation between RAD51 overexpression and poor prognosis nor publications describing the level of DNA damage in HR-inactivated cells (by siRNA and inhibitors or in BRCA-deficient cells).

Typos : phosphorylation sites of RPA (I assume it is Ser4/8) are not correctly indicated in Mat/Meth. “DNA” and not “DBA”, line 37. Previously, line 155. “poor” is repeated line 77

English is not my mother tongue so please apologize for this comment if I am wrong, but I think that few verbs/words are missing : line 37 (was), 39 (was), 179 (to), 289 (was), 292 (was)

There is no legend and mat/meth on the supplementary figures making them difficult to understand sometimes.

Author Response

POINT BY POINT RESPONSE TO REVIEWER 2 ATTACHED

Round 2

Reviewer 1 Report

The authors answered/filled in most of my comments. I think the manuscript can be accepted. 

Now, I suggest improving the quality of the figures. Some letters in Fig 2 are distorted or cut. Fig 3E, I would make with the same formatting as other graphs. In Fig 4, there are some unnecessary lines around some of the graphs. Graphs in Fig 5 are not aligned, and they are again a little bit distorted.

Author Response

Reviewer 1:

Comment 1: Overall, this could be a good work but I have to admit, I am not satisfied with the proofs and the conclusions made from them, Materials and methods section was disappointing. The whole manuscript is based on that elevated levels of RAD51 lead to DNA damage and poor prognosis. Authors in this study use cell lines with increased RAD51 levels. It would be good to validate these results with mild overexpression of RAD51 in some control cells (noncancerous/cancerous) and see if it is also associated with elevated DNA damage.

Response: In new Figure 3E, we now show that RAD51 overexpression in normal human fibroblast cells increases micronuclei (a marker of genomic instability) by ∼ 20-fold. These data are consistent with our previous data demonstrating that inhibition of RAD51 in cancer cells [(esophageal adenocarcinoma (EAC) and multiple myeloma (MM)], by transgenic and/or chemical modifications, inhibits HR and genomic instability9,10 and etoposide-induced DNA damage35. Text has been revised accordingly.

Comment 2: RAD51 is quite important protein its depletion has serious consequences, as altering the cell cycle progression. Accumulation in G1 or G2 phase could lower the DNA damage markers because most of the endogenous damage occurs during S-phase. I suggest to quantify the cell cycle after RAD51 depletion and inhibition. This is a major point for the review. If the negative effects of elevated RAD51 are truly because of hyperactive HR, maybe the effect of RAD51 depletion/inhibition could be mimicked by depletion/inhibition of another HR protein, which will not have such effect on cell cycle. Optionally, perform epistatic analysis between RAD51 and other HR protein.

Response: In cancers (such as EAC and MM), RAD51 is spontaneously elevated and dysregulated, contributing to unnecessary recombination events leading to genomic instability9,10, development of drug resistance10 as well as contributes to tumor growth11. Therefore, inhibition of RAD51, by transgenic and/or chemical modifications, inhibits HR and genomic instability9,10 as well as inhibits tumor growth with no toxicity to mice11.

            Although in normal cells, regulatory mechanisms ensure that RAD51 is active and HR takes place preferably in G2 phase of cell cycle16, this is not necessarily true for dysregulated HR in cancer situation. Data from our laboratory demonstrate that dysregulated RAD51 and HR are key contributors to increased spontaneous DNA damage. In addition to dysregulated HR, checkpoint controls are also dysregulated in cancer. Thus dysregulated, unscheduled and unnecessary recombination events and loss of checkpoint controls are attributed to genomic instability in cancer. Our published and unpublished data also show that besides RAD51, the suppression of other HR proteins APE1, RAD54B, FEN1 and EXO1 also inhibit DNA damage and genomic instability28,34. However, new Supplementary Figure 6 shows cell cycle data.

Comment 3: In this manuscript authors show that RAD51 inhibition/depletion rather decrease DNA damage with or without DNA damaging compounds. That is interesting, I would expect the opposite results, as it is shown in cited publication number 8, where both mild and severe RAD51 depletion elevated markers of DNA damage. Decreased levels of micronuclei could be explained by prevention of fork reversal and thus shifting the stalled replication fork into restart. Maybe this could be discussed in the discussion section. Cell cycle alteration could also explain lowered numbers of micronuclei as they are created after mitosis.

Response: In current manuscript the impact of RAD51 inhibition/depletion was strictly monitored in the live cell fraction. Recent unpublished and published data from our lab indicate that the treatments which inhibit RAD51 expression and/or HR activity inhibit DNA breaks (as assessed by g-H2AX) in the live cell fraction28,34-35. In cited publication number 8, the impact of RAD51 inhibition on DNA breaks was not strictly assessed in live cell fraction but entire cell population was investigated which included dead and dying cells which can have varying levels of DNA damage. Difference in the evaluation of overall cell population vs. live cell fraction has also been discussed in our recent publication28.

Comment 4: The immune dysregulation could be an interesting point, but I think it needs some additional work. Authors should test whether the cGAS-STING pathway is activated in the presence of elevated RAD51 levels and suppressed upon inhibition/depletion. cGAS-STING pathway is activated by extranuclear DNA, so should be activated in presence of micronuclei. This pathway could be responsible for the immune response. I believe this experiment could increase the impact of the publication.

Response: Our new data (new Figure 3E and new Supplementary Figure 5) demonstrate that RAD51 overexpression in normal human diploid fibroblast (HDF) cells caused a marked increase in genomic instability (as assessed from increase in micronuclei). However, no increase in cGAS expression was observed. In fact, cGAS expression was reduced following overexpression of RAD51 in normal human diploid fibroblasts. Consistently, the overexpression of RAD51 in breast cancer cells led to increase in DNA breaks and loss of cGAS expression (New Figure 2C). Overexpression of RAD51 in EAC (FLO-1) cells also led to slight loss of cGAS expression (new Supplementary Figure 5). Since cGAS-STING pathway is the cytosolic double-stranded DNA sensor, our data suggest that overexpression of RAD51 inhibited or reduced the release of DNA fragments (produced from rupture of micronuclei or other genomic rearrangements) into cytoplasm. It is possible that increased HR activity following RAD51 overexpression engaged these DNA/chromosomal fragments into aberrant recombination and other genomic events such as chromothripsis. Thus, elevated RAD51/HR-associated impact on immune system may involve mechanisms other than cGAS/STING (please see the review from our laboratory16). Text has been revised.

Comment 5: Materials and methods should be more descriptive. It is not necessary to describe the procedure but at least used methodology and basic facts should be stated. Sequence of shRNA and basic information about antibodies (company, type, catalogue number) should also listed. How quantification of Western blots was made? How many replicates were done? This is a major point for the review. Method of micronuclei evaluation refers to previous publication, however, in cited publication, there is again reference to another publication only to find out that it was performed by commercial kit. The reader must open two additional publications to see the name of the kit which could be stated in MM of this manuscript! I feel that the way it is currently in the manuscript is only to get more citations, which I do not consider very ethical.

Response: We have now provided more details in the Methods section including the information about shRNAs and antibodies. The citing of our own publications for the evaluation of micronuclei is to emphasize our specific approach of evaluation of micronuclei in live cell fraction only. To our knowledge, we are the first group to highlight that the impact of a treatment on genomic integrity cannot be accurately evaluated in the presence of dead cells28. We, therefore, remove dead cells and evaluate the genomic impact of inhibitor in cells which survive the treatment. This is extremely important because cell death/apoptosis is also associated with DNA breaks which can hinder/confuse the results during evaluation of the impact of treatment on genomic integrity/stability. This is the reason we wrote, “--as reported by us previously”. However, we have now revised the text.

Comment 6: Page 4, line 155 – Typo: previously

Response: This has been corrected.

Comment 7: Figure 2 – how long were the cells depleted? How long were the cells treated with RI-1? This is crucial information. The length of treatment/depletion will affect cell cycle. In Fig 3, there is the length of treatment. But is it the same as in figure 2?

Response: This information has been added to figure legends.

Comment 8: Figure 2 – how long was the treatment with CPT or 5-FU?

Response: This information has been added to figure legend.

Comment 9: Figure 2 BII – Normalized expression is not applicable for gH2AX. The same, p.13, line 380: pRPA32 is not overexpressed but phosphorylated

Response: Quantitations of all bands (including g-H2AX) are presented relative to GAPDH. The statement on p.13, line 380 has been revised.

Comment 10: Figure 2 BII – How the quantification was made? It should be at least from three experiments, and it should be shown with error bars.

Response: Quantitation was done using Image J software.